# High Hydrostatic Pressure Treatment Ensures the Microbiological Safety of Human Milk Including *Bacillus cereus* and Preservation of Bioactive Proteins Including Lipase and Immuno-Proteins: A Narrative Review

**DOI:** 10.3390/foods10061327

**Published:** 2021-06-09

**Authors:** Claude Billeaud

**Affiliations:** Neonatology & Nutrition, CIC Pédiatrique 1401, INSERM, Hopital des Enfants, CHU Pellegrin Place Amelie Raba Leon, 33076 Bordeaux, France; claude.billeaud@chu-bordeaux.fr; Tel.: +33-688-237-374

**Keywords:** HHP, breast milk, *Bacillus cereus*, human milk bank, lipase

## Abstract

Breast milk is the nutritional reference for the child and especially for the preterm infant. Breast milk is better than donated breast milk (DHM), but if breast milk is not available, DHM is distributed by the Human Milk Bank (HMB). Raw Human Milk is better than HMB milk, but it may contain dangerous germs, so it is usually milk pasteurized by a Holder treatment (62.5 °C 30 min). However, Holder does not destroy all germs, and in particular, in 7% to 14%, the spores of *Bacillus cereus* are found, and it also destroys the microbiota, lipase BSSL and immune proteins. Another technique, High-Temperature Short Time (HTST 72 °C, 5–15 s), has been tried, which is imperfect, does not destroy *Bacillus cereus*, but degrades the lipase and partially the immune proteins. Therefore, techniques that do not treat by temperature have been proposed. For more than 25 years, high hydrostatic pressure has been tried with pressures from 100 to 800 MPa. Pressures above 400 MPa can alter the immune proteins without destroying the *Bacillus cereus*. We propose a High Hydrostatic Pressure (HHP) with four pressure cycles ranging from 50–150 MPa to promote *Bacillus cereus* germination and a 350 MPa Pressure that destroys 10^6^ *Bacillus cereus* and retains 80–100% of lipase, lysozyme, lactoferrin and 64% of IgAs. Other HHP techniques are being tested. We propose a literature review of these techniques.

## 1. Introduction

Breast milk is the standard nutrient appropriate for infant development and especially for preterm. Breast milk is the best, but it is not always available, so breast milk provided by breast milk banks (HMB) is sometimes used [1]. In this study, we present the problem of contamination treatment to eliminate pathogen contamination. The optimal method eliminates all pathogens, including Bacillus cereus spores, and preserves the bioactive components of donated breast milk: fat absorption enhancing HM lipase and immune proteins (lysozyme, lactoferrin and IgAs).The main process used to pasteurize donated breast milk (DHM) is heat treatment, i.e., the Holder process—low-temperature long time (LTLT, 62.5 °C, 30 min) [2] and high-temperature short time (HTST, 72 °C, 5–15 s) and has been recently studied.

Neither of these processes destroys bacterial spores. In addition, LTLT alters the activity of many HM components (enzymatic, nutritional or immune components).

Over the past 25 years, high hydrostatic pressure has been developed primarily for food processing.

Different assays to apply high hydrostatic pressure (HHP) treatment to the decontamination of breast milk have been tested in recent years. Nevertheless, these studies were partial and did not lead to any specific process able to give an appropriate answer to the challenge (irreversible inactivation of all pathogens while preserving the activity of the component).

We found that an appropriate combination of different parameters characterizing the application of pressure (pressurization rate and latency period) at low-temperature (38 °C) could lead to the inactivation of both vegetative and spore forms [3]. In order to achieve conditions that preserve the maximum activity of the breast milk components, a temperature close to that of the human body was chosen. In this study, we provide a review of the properties of different methods as well as the properties of HHP.

## 2. Heat Treatment of Human Milk

### 2.1. Holder Pasteurization

Now, pasteurization of milk is an essential step to inactivate pathogens and part of the microbiota. It is the most commonly used pasteurization in HMBs LTLT, also known as Holder pasteurization (62°5–30′). However, this heat treatment can degrade biochemical components, such as IgAs, lactoferrin, some trace elements and several vitamins (Da Cost Evans [4]; Koenig [5]; It can also decrease the antioxidant capacity of milk, and it completely inactivates the lipase (Henderson [6]). Because of these losses, data supporting the use of pasteurized DHM as the sole diet for preterm, sick or preterm infants are incomplete. Cytokine concentrations also decrease after pasteurization, and there is a suggestion that more pro-inflammatory—rather than anti-inflammatory—cytokines. After LTLT pasteurization, there is a significant decrease in the concentrations of ascorbic, dehydroascorbic acid, α-tocopherol and γ-tocopherol. However, milk fatty acids, including long-chain polyunsaturated fatty acids, are not affected. Heat treatment of DHM implies a decrease in its antioxidant properties; however, if necessary, HTST should be the method of choice in terms of DHM oxidative status (Silvestre [7]). To improve the nutritional quality of processed donor HM, treatments other than Holder pasteurization need to be explored.

### 2.2. High-Temperature Short Time (HTST)

High-temperature processing (HTST) is another rapid heating strategy (72 °C × 5–15 s).

It is a method that seems better to LTLT (G Moro [8]). However, this process does not destroy spores, and the safety does not reach that of HPP, which is the only process that combines the safety of nutritional and biological quality of DHM. In addition, HTST requires a technological investment and is only available at the industrial level.

It has been reported that there is a 0–20% loss of sIgA, a 0–25% loss of total IgA, a 0–85% loss of lactoferrin and a total loss of lipase after STHT treatment (Goldblum [9]). There are variable effects of the HTST treatment on the lysozyme content of milk, ranging from a 20% to 40% loss. Chantry [10] showed that the IgA concentration measured after flash heating was almost identical to that reported by Goldblum [9] (a decrease in IgA concentration from 0.37 to 0.30 mg/mL) (19%) after heating breast milk to 72 °C for 15 s. Thus, all forms of heat treatment for the HM donor affect many milk components and, presumably, their efficacy. For this reason, treatment methods other than heat treatment should be explored.

## 3. High-Pressure Processing

### 3.1. High-Pressure Processing and Pathogens

#### 3.1.1. High-Pressure Processing (HPP)

HPP can be used as the non-thermal pasteurization of foods, and it can inactivate microorganisms (Balci [11], Chawla et al. [12], Rendueles et al. [13]), producing safe and minimally processed foods with satisfactory and organoleptic qualities.

High-pressure processing (HPP) (100–800 MPa) is one of the most promising methods, processing and preserving foods at room temperature (Cheftel [14]; San Martin [15]).

Research into the application of HPP for milk preservation began with Hite [16]. He extended it to milk and other food pressure treatment.

Advances in materials science and the metallurgical industries in the use of HPP techniques during the period 1955–1980 enabled industrial applications in the biosciences (Demazeau 2008 [17]).

In the mid-1980s, HP processes were renewed in Japan to ensure microbial safety of food (Hayashi [18]) in France (Demazeau and Rivalain [19]).

The first commercial products treated under HHP conditions appeared on the market in 1991. Today, thanks to the progress made in the knowledge of the behavior of food constituents under high-pressure conditions and in the development of high-pressure industrial equipment, HHP-treated food products are available in many countries and many products include plant (36%), meat (31%), juices (11%), seafood and fish (16%) and other products (6%).

#### 3.1.2. Comparison of HM Composition Treated by Pasteurization or High-Pressure

HPP uses water or a mixture of water and glycol as a means of transmitting pressures between 100 and 800 MPa to the milk (Trujillo [20]). Due to the low energy resulting from the application of HHP on liquid phases, only weak bonds can be affected. Therefore, small molecules, such as amino acids and vitamins, are not affected (Balci [11]; Lullien-Pellerin [21]). HPP can denature whey (Moatsou. [22]). However, Viazis, 2007 [23] demonstrated that HHP could better maintain IgA and lysozyme in HM than Holder pasteurization, and Permanyer [24] confirmed that IgAs, could be better maintained after HHP than after Holder pasteurization. Viazis, 2008 [25] was the first to show that HHP could inactivate five selected bacterial pathogens in breast milk. All these pioneers suggest that HHP may be a potential alternative to Holder pasteurization in HMB. However, we need studies on the effects of HHP on other components of DHM.

One study compared the effects of LTLT and HHP on fatty acid composition and on vitamin C and tocopherol in DHM (Moltó-Puigmartí [26]). In addition, Contador [27] and Delgado et al [28] varied the pressure (300, 600 and 900 MPa) and temperature (50, 65 and 80 °C) and studied the levels of tocopherols, fatty acids, cytokines, leukocytes and immunoglobulins (IgM, IgA and IgG) in the DHM. Treatment of breast milk at 65 °C and 80 °C, regardless of the pressure value, induced a decrease in α-tocopherol compared with untreated milk. The fatty acid content varied with the highest pressure values (600 and 900 MPa) and the highest temperatures (65 °C and 80 °C). On the other hand, HHP had a minimal effect on some cytokines, IL-2, IL-17 and IFN-γ. Leukocytes are highly affected by both HHP and thermal treatments. Only the treatment under 300 MPa and 50 °C maintained some immunoglobulin levels (IgM 75%, Ig 48% and IgG 100% retention).

Contador et al. [27] compared LTLT pasteurization (62.5 °C–30 min) to HHP (400 or 600 MPa for 3 or 6 min) on immune cells (leukocyte content) and immunoglobulin content (IgM, IgA and IgG). HHP at 400 MPa (for 3 or 6 min) better maintained the original immunoglobulin levels. HHP at 400 MPa appeared to preserve leukocytes more (10–14%) than LTLT. HPP would be better to maintain various components of DHM (compared with LTLT). **The lipase, particularly, is destroyed when it is over 40 °C and preserved even at 600 MPa. So, at 62.5° (LTLT), it is completely destroyed.**

In milk (Johnston [29]), the pressure-induced dissociation of colloidal calcium phosphate and the denaturation of milk serum proteins can change and/or improve its technological properties. In addition to microbial destruction, the effects of HHP on protein structure and mineral balance suggest (Devi [30]) different applications on the production of cheese and yogurt and the preparation of dairy products with new textures (Martínez-Rodríguez [31]). When milk is subjected to HHP, casein micelles are disintegrated into smaller particles (Johnston [29]). This results in an increase in caseins and whey proteins that become sedimentable by centrifugation and precipitable at pH 4.6. Law [32] reported that after pressure treatment up to 500 MPa at 25 °C, β-lactoglobulin is the most easily denatured serum protein. Denaturation of immunoglobulins and α-lactalbumin occurs only at the highest pressures and at temperatures from 50 °C.

An important study was devoted to the effects of HHP on the structure and dynamics of β-lactoglobulin (Russo [33]). In this study, the structural results of a high-pressure unfolding of β-lactoglobulin are observed above a value of 300 MPa I (Gosch [34]). The highly variable microbial quality of raw colostrum and its heat-sensitive nature complicate the preparation of safe colostrum products for human use. A new process combining microfiltration (1.4–0.8 μm) and HHP (400–500 MPa, 10 min) can lead to a significant microbial reduction. Calcium phosphate solubility decreases with increasing temperature and increases with pressure (Buchheim [35]).

Studies by Gervilla [36] on the free fatty acid (FFA) content of sheep′s milk showed that HHP at 100–500 MPa at 4, 25 and 50 °C did not increase lipolysis. Even some treatments at 50 °C showed a lower FFA content than fresh raw milk. This phenomenon avoids flavors derived from the lipolytic rancidity of milk. Hydrostatic pressure up to 500 MPa causes changes in the size and distribution of milk fat globules in sheep′s milk. HPP at 25 and 50 °C showed a tendency to increase the number of small globules in the 1–2 µm range, whereas at 4 °C this tendency was reversed, but no damage to the milk fat globule membrane occurred. In contrast, a study on goat milk revealed that lipoprotein lipase was somewhat resistant to pressurization (Buffa [37]). It should be noted that in human milk, there is not only lipoprotein lipase but also BSSL.

Lactose in milk can be isomerized into lactulose by heating, and then transformed into acids and sugars. On the other hand, HHP does not cause degradation of these compounds after pressurization (100–400 MPa; 10–60 min at 25 °C). No Maillard reaction or lactose isomerization occurs in milk after HHP (Lopez [38]). Milk with casein micelles of reduced diameter improves the coagulation properties. The disintegration of the micelles produced by HHP changes the color of the milk (Lopez [38]; Buffa [37]). There is a decrease in lightness (L*) and an increase in greenness (b*) and yellowness (Y*) of HPP-treated sheep milk (Gervilla [39]). The decrease in L* value due to the disintegration of casein micelles by HHP into small fragments increases the translucency of milk (Johnston [29]). This phenomenon was visually negligible in sheep’s milk, which is important to avoid rejection by potential consumers.

Similar results were found by Mussa and Ramaswamy [40] in cow’s milk, showing the low sensitivity of milk color to pressure. Warming samples from 4 to 43 °C caused the color values of HPP milk to return to values of untreated HM (Needs [41]).

While both covalent and non-covalent bonds are affected by temperature, HHP at ambient temperatures disrupts only weak bonds, so small molecules such as vitamins, amino acids, simple sugars and flavors are not altered by HHP (Cheftel [42]). The DHM treatment with HPP at 400 MPa does not result in loss of vitamins B1 and B6. Garcia-Grael [43] found that HPP at 400 MPa for 15 min at 40–60 °C reduced, and at 25–60 °C maintained the organoleptic properties of milk. So, this suggests that these treatments could produce milk with good sensory properties and increased preservation.

#### 3.1.3. HHP and Human Milk Safety

##### HHP and Bacteria

The resistance of pathogens to pressure in food is highly variable depending on the parameters of HHP (Pressure, Cycles, Temperature), the type of food, the properties and the physiological state of the pathogen (Smelt [44]). Exponentially growing microbes are more sensitive to pressure than stationary phase pathogens. Spores are always more resistant than vegetative bacteria and can survive a pressure of 1000 MPa. Germination of spores can be stimulated by pressures of 50 to 150 MPa. The vegetative forms can then be killed by heat or gentle pressure treatments. Gram-positive organisms tend to be more resistant to pressure than Gram-negative organisms. However, a considerable variation in pressure resistance within strains of the same species has been demonstrated in both Gram-positive and Gram-negative microorganisms. There is considerable variation in pressure resistance within strains of the same species. Gram-positives require HHP of 500–600 MPa at 25 °C for 10 min to be inactivated, whereas Gram-negatives are inactivated with pressures of 300–400 MPa at 25 °C for 10 min. Recently, a Gram-positive bacterium (*S. aureus*) was inactivated at 200 MPa in human blood plasma (Rivalain et al. [45]). Vegetative yeasts and molds are the most sensitive to pressure (Eperlan [44]). Numerous studies on the inactivation of pathogens in milk (naturally present or inoculated) by HPP have recently been performed and have shown that it is possible to obtain milk with a pressure of 400–600 MPa comparable to that of pasteurized milk (72 °C, 15 s), depending on the microbiological quality of the milk (Kolakowski [46], Mussa [41]), but with exceptions due to HPP-resistant spores.

In addition, the combined efficacy of HHP with mild temperatures (30–50 °C) and/or with bacteriocins (nisin, pediocin, lacticin, etc.) on foodborne bacteria and spores was tested. Some of these combinations significantly improve the efficacy of HPP, sometimes showing synergistic inactivation between HPP and natural antimicrobial substances (Garcıa Graells [43], Alpas [47]; Morgan [48]). Taking into account all the parameters of HP and temperature, we designed a specific high-pressure treatment of breast milk. The challenge was to inactivate the widest range of pathogens while preserving the activity of most of its major bioactive components in breast milk. We, therefore, developed this method, inoculating different germs in vegetative and/or spore form. As a result, in October 2012, we filed a French patent application (French patent 12.60215 HPBioTECH, filed 26 October 2012) by HPBioTECH and the Bordeaux University Hospital.

##### Effects on Bacteria and Yeasts

For the past 30 years, the pressure inactivation of microorganisms has been developed in biosciences, in particular for foods and more recently for biological products, including pharmaceuticals. In many past studies, the effect of HHP on pathogens focused mainly on the effect of an increase in the pressure value. To assure the safety of pharmaceutical products containing fragile therapeutic components, the development of new decontamination processes at minimal pressure values is needed to maintain their therapeutic properties (Rigaldie et al. [49]).

The effect of HPP parameters was evaluated on the inactivation of *Staphylococcus aureus* ATCC 6538, which is an opportunistic pathogen of important relevance in the medical, pharmaceutical and food domains.

##### Effects on Bacterial Spores

For 30 years, in many previous studies, the effect of HHP on pathogens focused mainly on the effect of a variation of HHP in food and fragile pharmaceutical products; the development of new processes at minimum pressure values, necessary to maintain their therapeutic properties (Reineke et al. [50,51]). The effect of HHP parameters was evaluated on the inactivation of *Staphylococcus aureus ATCC 6538*, which is important in the medical, pharmaceutical and food fields. Human blood plasma demonstrated a high inactivation rate (a 5 log10 decrease) at a pressure level as low as 200 MPa. Complete inactivation was achieved under the following conditions: PR = 50 MPa, AM = 5 × 2 min, T = −5 °C and p = 300 MPa.

##### Effects on Viruses

There is great diversity within the virus family (Norton [52]). They consist of a protein envelope, called a capsid, composed of a number of proteins (capsomers) that contain a central nucleic acid core. They may also contain a small number of enzymes necessary for their infectivity. The mechanisms of viral inactivation by HHP are related to dissociation and/or denaturation of the virus envelope (Silva et al. [53]. Even in the case of enveloped viruses, the envelope can be denatured (Nakagami [54]).It can be a very small alteration of capsid proteins (Kingsley [55]) or receptor recognition proteins (Pontes et al. [56]), which leads to a loss of infectivity. The range of structural diversity involved is reflected in a wide diversity of pressure resistance (Smelt 1998 [45]). The most common human viruses are Norwalk type virus (SRSV), hepatitis A, rotavirus and astrovirus. Inactivation of feline calicivirus, adenovirus and hepatitis A can be achieved by treatment at 275 MPa for 5 min, 400 MPa for 15 min and 450 MPa for 5 min (Kingsley, 2002 [55]). A few studies have shown remarkable survival of poliovirus (Nakagami [54]). Foot-and-mouth disease virus was reduced by 102.9 units per treatment at 220 MPa for 1 h (Kingsley et al. [55]). The mode of inactivation of viruses by HHP has not been fully explained, although the viral envelope seems to be a target for HHP inactivation. Electron microscopy showed that high pressure at 300 MPa damaged the virus envelope and prevented virus particles from binding to cells. HPP holds promise for the inactivation of HSV-1, cytomegalovirus (Landolfo [57]) and other enveloped viruses (Nakagami [54]). Pressure-induced dissociation can be fully reversible or irreversible. HHP can also induce minor changes in viral structures without disassembling the whole particle (Gaspar [58]) but without infectivity.

## 4. Comparison of Significant Different HHP in Development

### 4.1. The High Hydrostatic Pressure Process Developed by HPBioTECH & CHU Bordeaux

Demazeau G. [3] used a pressure value of 350 MPa, a pressurization rate PR of 1 MPa/s and a cyclic application mode MA (3 or 4 cycles with a plateau of 5 min for each cycle) combined with a latency period of 5 min and a temperature of 38 °C.

This equipment was used to set up the HHP treatment of human milk. The challenge was to combine all the parameters governing HHP in order to reach the highest microbial reduction (including bacterial spores) and the optimal preservation of the main components of human milk.

We first carried out a pilot study that analyzed the reproducibility of HHP, compared to LTLT and raw milk of the same mother, in a bacteriological and biological study.

#### 4.1.1. Inactivation of All Pathogens

Efficacy on the total vegetative flora (rejected because of its important initial contamination, up to 10^8^ cfu/mL). Efficacy on *Staphylococcus aureus*: total inactivation for an initial concentration was between 10^6^ and 10^7^ cfu/mL. Efficacy on bacterial spores: the HHP process was evaluated in HM inoculated by bacterial spores (either *Bacillus subtilis* or *Bacillus cereus*). This process was able to inactivate up to 10^6^ cfu/mL for both species. Efficacy on viruses: the HHP process was tested on cytomegalovirus. Complete inactivation of CMV was observed with an initial viral particle content greater than 7 logs in breast milk after HHP application.. The HHP process irreversibly inactivates all pathogens, and the HHP-treated HM is microbiologically stable for a period of at least six months at a storage temperature of -18 °C. HM maintain complete sterility and lipase.

#### 4.1.2. Preservation of the HM Bioactive Components

##### Lipids

Lipase: The application of this HHP process led to the preservation of 88% of the lipase activity, while it was completely destroyed by LTLT pasteurization.

MFG (Milk Fat Globules) granulometry: In terms of size, the MFG population was of a bimodal type with a mean diameter (d4.3) MFG for milk (raw milk: 5.5 µm; LTLT: 5.6 µm; HHP: 5.4 µm). In addition, the proportion of “small” MFG was greater in raw milk and HHP milk (d3.2 = 0.6 vs. 0.8 µm, respectively) than in LTLT breast milk (d3.2 = 3.1 µm), whereas LTLT favors coalescence and thus the amount of “large” MFG.

Animal pilot study: the intestinal absorption of lipids in breast milk was measured in lymphatic duct cannulated rats fed either raw milk, LTLT milk or HHP milk. The first data acquired on a limited number of animals (n = 3/group) show that the absorption of lipids in rats receiving HHP milk tends (*p* < 0.3) to be improved (7.1 mg/mL lymph) compared to animals receiving LTLT milk (5.9 mg/mL lymph). However, these results are only primary and do not show a significant difference.

##### Protein Metabolism

This method of HHP preserves IgAs 64%, lysozyme 95%, lactoferrin 100%, caseins was close to 80%, α-lactalbumin: 90%.

Cytokines: the highest concentrations of inflammatory cytokines were observed in pasteurized samples compared to the raw and high hydrostatic milk breast milk samples. This difference is probably related to the highest rate of cytolysis during LTLT pasteurization when compared to HHP.

### 4.2. Other Studies with HHP

Raso [59] studied the different methods of germination of *Bacillus cereus* spores and therefore used a 250 MPa HHP cycle, added alanine to the milk and then completed a 690 MPa cycle. Under these conditions, substantial protein modification and probably lipase modification were achieved. This shows that it is difficult to destroy *Bacillus cereus* with a single cycle of HHP and certainly not below 600 MPa and at a temperature below 40 °C, as in our study, in which *Bacillus cereus* was germinated by an initial cycle at 50–150 MPa and with a temperature of 38 °C.

Jarzynka [60] treated breast milk with HHP at 450 MPa for 15 min, but 10^2^ *Bacillus cereus* remained after treatment. She preserved the breast milk by lyophilization. For the 10 years that we have been using this process in Marmande as a method of conservation, we have not known how the *Bacillus cereus* remaining after HHP can be sterilized. Moreover, the cost of this treatment would require the cost of a Pascalizator and a freeze-drying machine, i.e., a total of one million euros.

Pitino [61] compared HHP at 500 MPa for 8 min, UV-C at 250 nm for 25 min with flash heating and Holder’s method. He measured the bacteriology of breast milk, considering milk with less than 10^3^ edible germs, which would not be considered sterile. He measured lipase, lactoferrin and lysozyme, which in our procedure, [3] were, respectively, 88%, 100% and 95%. Lipase was absent with LTLT and flash heating, reduced by 48% with UV-C and maintained with HHP. Lysozyme was reduced by 44% with flash heating, reduced by 74% with UV-C and, strangely, was normal with Holder and with HHP, as per our results. Lactoferrin was decreased in all methods: substantially by flash heating (74%) and UV-C (48%) and by 25% by HHP. Lactoferrin was present at 100%. This study has the limitation of not having made inoculations of *Bacillus cereus* and admitting 10^3^ germs considered as normal values, while in France and most HMB we require <1 cfu. However, among the other methods, UV-C (Christen [62]) and flash heating are disqualified by this study, and HHP preserves the HM (Table 1).

Malinowska-Pańczyk [63] worked on a method with very low HHP pressure (60 to 190 MPa) and a milk temperature < 0 °C. The pressure treatment at 193 MPa and −20 °C on membrane damage bound ATPases and degradation of nucleic acids of E.Coli. However, she did not do the *Bacillus cereus* challenge test.

## 5. Conclusions

Holder’s pasteurization method, currently used by all HMBs, has many bacteriological defects. It does not destroy *Bacillus cereus*, which can cause severe preterm infections (Rigourd [64]), and does not preserve lipase, lysozyme, lactoferrin or IgAs (Contador [27]). This is the most common method in HMBs worldwide for treating donated maternal milk. It is the most expensive method if we consider that between 7% and 14% (or 1000 L) of the milk in France (in an HMB such as in Bordeaux-Marmande or Lyon, which collects more than 10,000 L of milk) is discarded due to the presence of *Bacillus cereus*. At a rate of 150 €/L, this costs 150,000 € of losses each year. HHP is the best technique for ensuring sterility when we know that the price of a Pascalizator costs 300,000 €. For over two years, it has amortized the price of the device. Researchers must study UV-C, Ultrasounds and Cavitation. Using HHP without exceeding 350 MPa and a temperature at 38 °C allows one to retain almost 100% of the BSSL lipase, which improves the absorption of not only fats, as demonstrated in our pilot study in rats, but also immune proteins (lactoferrin: 100%; lysozyme: 95%; iGas: 64%). In this study, HHP destroyed 10^6^ specimens of *Bacillus cereus* and *Staphylococcus aureus* and insures safety Human Milk.

## Figures and Tables

**Table 1 foods-10-01327-t001:** Comparison of HHP, LTLT, HTST and UV-C expressed in percentage of raw Human Milk. HHP preserve Biocomponent of HM: Lipase, lactoferrin, lysozyme and IgAs, but Demazeau HHP is the lower pressure with a maximum of 350 MPa and four cycles, one between 50–150 MPa to allow the germination of *Bacillus cereus Spores* and the last cycle at 350 MPa destroyed the vegetative form; Pitino considered that <10^3^ cfu is normal, but we consider that only <1 cfu/mL is sterile. Also, he did not do a challenge test with more than 10^6^ of *Bacillus cereus Spores*, and we say that with only one cycle, we can destroy it, even with one cycle of HHP > 600 mPa.

	Demazeau HHP	Buffin (LTLT)	MORO (HTST)	Christen (UV-C)	Pitino (HHP)
HHP	yes	no	no	no	yes
max MPa	350 × 15′				500 × 8′
Temperature	38 °C	62.5 °C × 30′	72 °C × 5–15′’		4 °C
UV-C	no	no	no	yes	
Lipase	79–100%	0	0	52%	80–100%
Lactoferrin	100%	21%	25%	40%	70%
Lysozyme	95%	84%	66%	30%	100%
IGas	64%	71%	nd	nd	nd
Bacteriology: cfu/mL	<1 cfu/mL Sterile	<1 cfu/mL sterile	<1 cfu/mL sterile		<10^3^ cfu/mL No sterile
*Bacillus cereus*	Destroyed	No Destroyed	No Destroyed	No Destroyed	No Destroyed

## Data Availability

The data details are displayed in the Publication of the studies in the reference [3,61,62] and are summarized in this review paper.

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
