# Peer review of "High Hydrostatic Pressure Treatment Ensures the Microbiological Safety of Human Milk Including Bacillus cereus and Preservation of Bioactive Proteins Including Lipase and Immuno-Proteins: A Narrative Review"

_foods, 2021, doi:10.3390/foods10061327_

Round 1
Reviewer 1 Report
Claude Billeaud’s interesting manuscript “High Hydrostatic Pressure Treatment Must Insure the Pathogens Safety of Human Milk Including Bacillus cereus and the Preservation of Bioactive Proteins Particularly Lipase and Immune Proteins: A Review”presents the various methods of decontaminating human milk of potential pathogens.
The article highlighted his innovative research on High Hydrostatic Pressure (HHP) treatment of human milk and the safety offered as well as beneficial preservation of key components. HHP was mainly compared to Holder Pasteurization and Short time high-temperature methods.
Major Revisions should include the following:
1)Organizing this article as a review and highlighting the HHP as one method of human milk treatment. The current structure of the article is confusing since it mixes a review style with presentation of novel results of HHP along with extensive details (such as on page 7 of 12, end of second paragraph which includes PR=50MPa, AM=5*2 min, T=-5ËšC, and p = 300MPa) which are not necessary to include in the context of a review. Similarly, page 3 of 12, section 3.1.2 under “Comparison of HM Composition Treated by Pasteurization or High Pressure,” includes technical details that are distracting and more appropriate for original research manuscript. Likewise, page 6 of 12, first paragraph under “HHP and Bacteria” is too long and detailed. I would advise to shorten these 2 specific paragraphs by eliminating excess details.
2)Recommend removing certain points that are not helpful to the flow of the manuscript.
-Page 3 of 12, Last sentence in last paragraph under section 3.1.1 “particularly RTE (ready to eat products).
-Page 5 of 12 “UHT milk” not sure what that is and it is not explained
-Page 8 of 12, section 4.2, last point which is “HM maintain complete sterility and lipase”, would remove lipase since it is discussed in the next section, 4.3 under Lipids.
-Page 10 of 12, Malinowska-Panczyk’s unpublished work is just mentioned but does not add any results, thus should be removed. Also, mentioning that “We have awaited her results since 2017”, may be added for clarity but leaves the impression that their work is incomplete and thus is not appropriate to state. Please remove this statement.
3)Important point to consider-Bacillus cereus has caused infections in neonates, but exact source is rarely identified, and thus far has not been traced back to pasteurized human milk from a milk bank. Thus, the milk is a theoretical, but in fact potential, source of B. cereus. May want to mention this in the conclusion.
4)It would be informative to see a table with human milk components compared between raw, LTLT, and HHP (i.e. Lipids, proteins and immune components)
Small revision points:
-IGas was very confusing, until I realized it was meant to be “IgAs”. Please correct these in the text.
-Please spell out all abbreviations. HHP and HPP are both used in the manuscript but both need to be spelled out the first time. Also, HP is used but not spelled out, does it mean Hydrostatic Pressure? HM presumably means human milk?
-Title needs editing: Insure to Ensure; Immune is misspelled and appears as “Immun”. Please correct.
Author Response
Answer to Reviewer 1:
I have answered all the points you have reported.
I have added a table as advised and I thank you for the suggestions that helped me to improve my manuscript
For point 1:
From point 4.1 to the conclusion, we have seen fit to detail the 4 most relevant studies
This gives the impression of an article, but these are the studies that tested the different HHP on both bacterio-logical results and bacteriology. Regarding the parameters of HHP, we can not only consider the Pressure but also the other parameters: number of cycles, duration of cycles, temperatures to have the desired effect on the bacterio-logical sterilization and preservation of bioactive components of breast milk
For point 2
We have reduced the details on page 3,5,7 and deleted the recommended details which however had their interest to be complete.
For page 8: we have deleted the repetition of the lipase which is included in the lipids section
For page 10: We have given more details on the data of Malinowska-Panczyck but in our opinion gives partial results on E.Coli and not other germs, which does not speak of the results of biological parameters.
For point 3: If the Bacillus Cereus does not necessarily come from breast milk, it is found in 7 to 14% after classic pasteurization and we must eliminate the contaminated batches (Rigourd). It is therefore essential to have a pasteurization process that gives a perfect sterility (<1cfu/ml)
For follow your advice we have made a table which summarize the Biologics and Bacteriologics results with the different authors
Reviewer 2 Report
The review deals with a current topic and describes the HHP technique with a rich and complete bibliography.
The bibliography is updated; the introduction is clear and comprehensive.
The comparison between traditional techniques and new technology is interesting and the studies on HHp appear complete and updated.
The article emphasized the HHP treatment of human milk in order to save economic waste. The Author compares the traditional technique vs the HHP treatment with an abundant and updated supporting bibliography. The article is well written and structured; the review of the pre-existing bibliography is clear and ready for use. The description of the HHP treatment is complete, comparing different parameters of pressure and time with a risk / benefit ratio. I believe that the review is useful for those who will read it.
no further comment
Author Response
Thank you for your gentle comments.